# COVID-19 Clusters in Belgian Nursing Homes: Impact of Facility Characteristics and Vaccination on Cluster Occurrence, Duration and Severity

**DOI:** 10.3390/v15010232

**Published:** 2023-01-13

**Authors:** Sara Dequeker, Milena Callies, Lucy Catteau, Laura Int Panis, Esma Islamaj, Sofieke Klamer, Katrien Latour, Marijke Pauwels, Catharina Vernemmen, Romain Mahieu, Hanna Masson, Muhammet Savsin, Etienne De Clercq, Mireille Thomas, Boudewijn Catry, Eline Vandael

**Affiliations:** 1Department of Epidemiology and Public Health, Sciensano, 1050 Brussels, Belgium; 2Department of Infectious Disease Prevention and Control, Common Community Commission, Brussels-Capital Region, 1040 Brussels, Belgium; 3Agency for Care and Health, Infection Prevention and Control, Government of Flanders, 1000 Brussels, Belgium; 4Direction de la Recherche, de la Statistique et de la Veille des Politiques, AVIQ, 6061 Charleroi, Belgium; 5Iriscare, Brussels Public Agency for Health and Social Care, 1040 Brussels, Belgium; 6Cellule des Maladies Infectieuses, Département Santé et Personnes Âgées, Ministère de la Communauté Germanophone, 4700 Eupen, Belgium; 7Faculty of Medecine, Université Libre de Bruxelles (ULB), 1050 Brussels, Belgium

**Keywords:** long-term care facilities, SARS-CoV-2, clusters, vaccination, infection prevention

## Abstract

COVID-19 severely affected nursing home residents from March 2020 onwards in Belgium. This study aimed to model the impact of vaccination and facility characteristics on cluster occurrence, duration and severity in this setting. Possible clusters were identified between June 2020 and January 2022, based on the Belgian COVID-19 surveillance in nursing homes. Median attack rates (AR) among residents and staff, case hospitalization rates (CHR) and case fatality rates (CFR) were calculated. A negative binomial model was used to identify the association between nursing home characteristics and the number of cases, hospital admissions and deaths and the duration of the cluster. A total of 2239 clusters were detected in more than 80% of nursing homes. Most of these (62%) occurred before the start of COVID-19 vaccination (end of December 2020). After vaccination, the number of clusters, the AR among residents and staff, the CHR and the CFR dropped. Previous cluster(s) and vaccination decreased the number of cases, hospital admissions and deaths among residents. Previous cluster experience and having started vaccination were protective factors. We recommend continued implementation of targeted interventions such as vaccination, large-scale screening and immediate implementation of additional infection prevention and control measures.

## 1. Introduction

The SARS-CoV-2 (severe acute respiratory syndrome coronavirus-2) pandemic severely affected the elderly population and people with chronic medical conditions, putting nursing home residents in a vulnerable position [1,2]. In Belgium, a national lockdown was introduced in March 2020, which included visitors no longer being allowed into nursing homes. In the meantime, a surveillance among nursing home residents and staff was set up in collaboration with the Belgian regional health authorities and Sciensano, the Belgian Scientific Institute of Public Health [1,3]. This surveillance allowed one to follow the evolution of the SARS-CoV-2 pandemic in the population residing in nursing homes and to act rapidly on signs of clustered infections.

Prior to 2022, the SARS-CoV-2 pandemic caused five successive waves of high numbers of COVID-19 cases in Belgium. A first wave occurred from 1 March to 21 June 2020, followed shortly by a second wave (31 August 2020 to 14 February 2021). During these first two waves, 12,447 nursing home residents died from SARS-CoV-2 infection, accounting for 57% of all COVID-19-related reported deaths in Belgium [4]. Strict infection prevention and control (IPC) measures of various types were implemented (e.g., visit restrictions, quarantine measures, cohorting, use of additional personal protective equipment (PPE), preventive and rapid testing) at regional and local levels from the onset of the pandemic in these nursing homes [1]. The Belgian national COVID-19 vaccination campaign was initiated in December 2020, with priority given to nursing home residents and staff. By the end of March 2021, 89% of residents and 77% of staff were fully vaccinated with a primary course of the BNT162b2 vaccine (Comirnaty, Pfizer-BioNTech) [5]. After this campaign, from the end of February 2021, some IPC measures implemented in nursing homes were relaxed (such as visitor restrictions, mask wearing for visitors) [6].

The second wave was followed by a third wave (15 February to 27 June 2021), which mostly affected the general population [6,7]. Meanwhile, the alpha variant (B.1.1.7) became dominant in February 2021. In early October, a fourth wave started to affect Belgium, with the delta variant (B.1.617.2) as the dominant variant. At the same time, a booster vaccination campaign with a mRNA COVID-19 vaccine (BNT162b2 or mRNA-1273 (Spikevax, Moderna)) began in nursing homes. Starting from 27 December 2021, the omicron variant (B.1.1.529) became dominant and triggered a fifth wave. Each successive variant showed different characteristics associated with a growth advantage: altered transmissibility, only partial immunologic cross-reactivity, altered population immunity and behaviour [8,9,10]. Moreover, a reduction in vaccine efficacy against emerging variants has been described, but vaccines still provide considerable protection and reduce disease severity [6,8].

During a pandemic, cluster detection in nursing homes is essential to assess the extent of disease spread and to properly plan for additional equipment and/or staff needs. When cluster infections are identified in time, loss of resources and lives are limited, as well as fear and panic in the general population [11].

In Belgium, COVID-19-positive cases and clusters within nursing homes and other residential institutions were to be declared to the responsible regional health authority [12]. Despite the legal obligation to report, it was challenging to estimate and interpret the number of clustered cases in nursing homes. Due to the high workload of nursing home staff and public health, and the large number of clusters, not all clusters were reported, or the regional health authorities decided to focus only on large outbreaks. Furthermore, the reporting method of these clusters differed between regional health authorities. In order to be able to follow the trend of clusters in these nursing homes, a cluster detection system based on the COVID-19 surveillance data was set up to identify possible clusters, which was used as an indicator to assess the extent of the COVID-19 epidemic in nursing homes.

The aim of this study is to describe the magnitude and severity of possible clusters in Belgian nursing homes between 22 June 2020 and 2 January 2022, and to determine an association with nursing home characteristics. In addition, the impact of vaccination on the occurrence of clusters in these facilities is explored.

## 2. Materials and Methods

### 2.1. Study Design

A national COVID-19 surveillance was set up in Belgian long-term care facilities in March 2020. The set-up and methodology of this surveillance has been described in a protocol [13] and by Vandael et al. [1]. The current manuscript focuses on long-term care facilities dedicated to elderly, namely nursing homes. All Belgian nursing homes (*n* = 1542) were invited to participate in this surveillance on a weekly basis. The collected data were aggregated at institutional level. In addition, individual data of deceased residents were collected.

### 2.2. Case Definition

The following definitions were applied [1,13,14]:
A confirmed case of a COVID-19 infection is a person with laboratory confirmation of the virus causing COVID-19 infection via a molecular test (PCR or rapid antigen test), irrespective of clinical signs and symptoms.A possible case of a COVID-19 infection is a person:with at least one of the following main symptoms of acute viral infection: cough, dyspnea; thoracic pain, acute anosmia (loss of the sense of smell) or dysgeusia without obvious cause;ORwith at least two of the following symptoms: fever, muscle pain, fatigue, rhinitis, sore throat, headache, anorexia, watery diarrhea with no obvious cause, acute confusion, sudden fall with no obvious cause;ORwith worsening of chronic respiratory symptoms (chronic obstructive pulmonary disease, asthma, chronic cough…);ORwho did not have a laboratory test or whose laboratory test is negative but who was diagnosed with COVID-19 based on a suggestive clinical presentation and a compatible computed tomography (CT) thorax.A confirmed COVID-19-related death is a resident who died due to a confirmed COVID-19 infection.A possible COVID-19-related death is a resident who died due to a possible COVID-19 infection.

### 2.3. Study Period

In the beginning of the pandemic, testing was prioritized in hospitals. Hence, nursing homes mostly reported possible cases. Starting in April/May 2020, cases could also be confirmed in nursing homes [15]. Because the cluster detection system is based on confirmed cases only, this manuscript focuses on data from 22 June 2020 (start of the first interwave period) until, and including, 2 January 2022. An overview of the number of cases, the general schedule of the vaccination campaigns in nursing homes and consecutive variants can be found in Figure 1.

### 2.4. Population

Nursing home residents included in this surveillance were persons at least staying more than one night in the institution. Nursing home staff are all personnel working in the facility (nursing staff, paramedical staff, animation team, staff concerned with cleaning, maintenance or quality control and long-term care facility managers and their administrative staff), not including staff who are absent for longer than one month, volunteers and students.

### 2.5. Data Processing, Analysis and Validation

The COVID-19 surveillance in Belgian NHs was organized by the regional health authorities. The raw data of all regions were merged in one database and analysed at national level by Sciensano, the Belgian institute for health [1,3]. More details of the different regional data collection tools and methods can be found in Vandael et al. [1] and Renard et al. [3]. The data extraction process and the data cleaning were automated as much as possible in SAS Enterprise Guide version 7.1. to avoid manual mistakes. More information about the automatic data cleaning and the used validation rules can be found in Vandael et al. [1].

The definitions applied to calculate possible clusters are presented in Table 1.

For each cluster, variables (characteristics of the institution, data about the current cluster, number of clusters that occurred within the institution within the study period, data about the vaccination campaign of the primary course, previous cluster) were calculated based on the COVID-19 nursing home surveillance data. A summary of those variables can be found in Appendix A. This process was automated as much as possible in SAS Enterprise Guide version 7.1, to avoid manual mistakes.

For the analyses, the database version of 9 February 2022 was used. Automatic data cleaning was performed on the surveillance data using different validation rules. Aberrant values were put as missing in the database [1]. Based on the surveillance data, a cluster database was built. If the number of infected residents within the cluster was higher than the number of residents, or if the number of infected staff members within the cluster was higher than the number of staff members, the cluster was excluded from the analyses (*n* = 70 or 3%).

Weeks are always displayed from Monday to Sunday. The bed occupation (reported by the nursing home) was used as denominator for the total number of residents (or if not available the total number of beds within the nursing home). For staff, a fixed self-reported number at the beginning of the crisis (first entry between March–May 2020) was used.

To calculate the incidence of confirmed COVID-19 cases, the sum of all newly reported confirmed COVID-19 cases through the surveillance was calculated per week. In order to calculate the incidence of confirmed COVID-19 cases within a cluster, a variable to indicate if the nursing home had an ongoing cluster on the reference date, was calculated and added to the surveillance database.

Analyses were performed using SAS Enterprise Guide version 7.1. Cumulative numbers, percentages and medians with interquartile range (IQR) were calculated where appropriate. Median attack rates (AR) among residents and staff, median case hospitalization rates (CHR) and median case fatality rates (CFR) were calculated. All participating nursing homes were divided into small and large institutions based on the median of the number of beds (small <P50, large >P50). Cluster characteristics (size (number of cases), number of hospital admissions and number of deaths within a cluster as well as cluster duration (days)) were compared by nursing home features (ownership (public, private for-profit institution, private nonprofit institution), size (small/large), region, presence of a previous cluster (yes/no), vaccination status (not started, first dose of the primary course given, second dose of the primary course given) and vaccination rate (≥80%, following the recommendations of the European Commission [5]) using the Wilcoxon–Mann–Whitney test, or the Kruskal–Wallis test in case k > 2. *p*-values ≤ 0.05 were considered statistically significant.

Because of the presence of overdispersion, a negative binomial model was used to identify association between the size of the nursing home, nursing home ownership, region, COVID-19 wave, history of a previous cluster and vaccination status and the dependent variables number of cases, hospital admissions, deaths and the number of days a cluster lasted. Associations were first assessed in univariate models. Independent variables significant at the 0.10 level were included in the final model. The variable ‘COVID-19 wave’ was excluded from the model because of the high correlation with ‘vaccination status’. A backward stepwise model selection was performed, comparing the Akaike information criterion (AIC) of each model, to determine the goodness of fit. The ‘vaccination rate’ among staff or residents was not included to avoid excluding all clusters occurring before the start of the vaccination campaign and because of the high correlation with the variables “COVID-19 wave” and “vaccination status”.

### 2.6. Ethical Considerations

The study protocol was approved by the ethical committee of the University of Ghent (23/10/2020, BC-08065). Informed consent was not required. Oral approval of the information safety commission was obtained.

## 3. Results

### 3.1. Participating Nursing Homes

On 22 June 2020, 1533 (99%) nursing homes accounting for more than 147,634 beds and 107,795 staff members (data missing for 5 nursing homes), participated in the surveillance. The median number of nursing home beds was 91 (IQR: 65–120), and the median number of nursing home staff members was 63 (IQR: 38–92). Overall, the bed occupation (calculated for nursing homes of which the number of beds and the actual number of residents was available) decreased from 98% in the beginning of April 2020 to 95% at the end of December 2021.

The number of beds (nursing home size) and the number of nursing homes differed between the three regional health authorities (Table 2).

The median weekly participation rate in the complete study period was 90% (IQR: 73–95%) with a maximum participation rate of 97% (week 46 in 2020: 9 November 2020–15 November 2020) and a minimum participation rate of 62% (week 52 in 2021: 27 December 2021–2 January 2022). The median participation rate was comparable for Flanders (94%, IQR:79–97%) and Wallonia (94%, IQR:71–97%), but lower for Brussels (76%, IQR: 67–81%).

### 3.2. Clusters

The median number of clusters occurring in one nursing home during the entire study period was 2 (IQR: 1–3) with a maximum of 7. For 19% of the participating nursing homes, no clusters were detected during the study period. The majority of the clusters (*n* = 1381; 62%) were detected before the start of the first COVID-19 vaccination campaign (primary course). In total, 102 clusters (4.6% of all clusters) occurred between the first and second dose of the primary vaccination schedule and 754 clusters (33.7%) were detected after the start of the second dose of the primary vaccination schedule. During that period, a total of 2239 clusters were detected in 1236 nursing homes.

Figure 2 shows the evolution of size of the COVID-19 clusters (number of confirmed COVID-19 cases) detected among nursing home residents between 22 June 2020 and 2 January 2022. The majority of large clusters (≥10 confirmed cases among residents only) were detected before the COVID-19 vaccination campaign. Large clusters accounted for 55% of all clusters before the start of the primary vaccination course. After the first dose and second dose of this primary course, large clusters accounted for 38% and 27%, respectively.

Table 3 gives an overview of the cluster characteristics (size (number of COVID-19 confirmed cases), duration (days)), number of hospital admissions and deaths and vaccination period between 22 June 2020 and 2 January 2022 for nursing home residents and/or staff. The median size of a cluster (number of cases) among residents and staff members was 11 (IQR: 4–30) during the entire study period. The median number of hospital admissions and deaths among residents due to a confirmed COVID-19 infection within a cluster during that same period was 0 (IQR: 0–2, 0–3, respectively).

### 3.3. Factors Associated with Cluster Emergence in Nursing Homes

The size of clusters occurring in nursing homes was significantly influenced by several factors, namely the region where the nursing home was localized (*p* = 0.03), the wave in which the cluster occurred (*p* < 0.001), the vaccination status of the institution (*p* < 0.001) and whether this was the first cluster occurring in the nursing home or not (*p* < 0.001). A cluster lasted significantly longer when the size of the nursing home was larger (*p* = 0.004), during the second wave and the fifth wave (*p* < 0.001), when this was the first cluster for the nursing home (*p* = 0.02) or when vaccination had not yet started within the nursing home (*p* < 0.001) (Table 4).

A vaccination rate among residents (two doses of the primary course) that reached the threshold as advised by the European commission (≥80%) significantly influenced the number of cases among residents (*p* = 0.04). Analogously, a vaccination rate among staff (two doses of the primary course) that reached the advice of the European commission (≥80%) had a significant influence on the number of hospital admissions among residents (*p* = 0.03) within a cluster occurring after the start of the vaccination campaign (Table 4).

A univariate and multivariate analysis was performed, the former can be found in Appendix A and the latter in Table 5. Based on the multivariate analysis, large nursing homes (incidence rate ratio, IRR: 1.26; 95% confidence interval, CI: 1.15–1.37) appear to experience larger clusters than other nursing homes. Residential institutions that had a previous cluster (IRR: 0.71; 95% CI: 0.63–0.81) and where the vaccination campaign had begun (first dose of primary course: IRR: 0.58; 95% CI: 0.47–0.72; second dose of primary course: IRR: 0.40; 95% CI: 0.36–0.44) were more likely to have smaller subsequent clusters (Table 5).

Higher numbers of COVID-19-related hospital admissions and deaths were observed in nursing homes located in the Wallonia region (IRR: 1.37; 95% CI: 1.17–1.60) and 1.41; 95% CI: 1.19–1.67, respectively) compared to nursing homes in other regions. The number of COVID-19 hospital admissions and deaths among nursing home residents within a cluster was influenced downwards by a previous cluster (IRR: 0.70; 95% CI: 0.57–0.86/0.67; 95% CI: 0.54–0.84) and the start of the vaccination campaign (first dose of primary course: IRR: 0.48; 95% CI: 0.33–0.70 0.47; 95% CI: 0.33–0.69; second dose of primary course: IRR: 0.26; 95% CI: 0.22–0.31/0.26; 95% CI: 0.22–0.31), respectively (Table 5).

Larger nursing homes (IRR: 1.13; 95% CI: 1.07–1.20) and facilities that already started with the second dose of the primary course (IRR: 1.32; 95% CI: 1.24–1.40) had a higher chance to have a long-lasting cluster compared to other nursing homes. Private nursing homes (both for profit; IRR: 0.93; 95% CI: 0.86–0.99 and nonprofit; IRR: 0.91; 95% CI: 0.85–0.97) and nursing homes where the first dose of the primary course was given (IRR: 0.81; 95% CI: 0.71–0.93) were more likely to have a cluster that was shorter compared to other nursing homes (Table 5).

## 4. Discussion

### 4.1. Main Results

While the cluster detection system was intended in the first place to estimate the trend of clusters in long-term care facilities, and especially nursing homes, to manage the crisis and mitigate the pandemic, here we explored a limited number of institutional characteristics that may be associated with the cluster size (in term of number of cases, hospital admissions, or deaths among residents) or the cluster duration. A total of 2239 clusters were detected in more than 80% of Belgian nursing homes between 22 June 2020 and 2 January 2022. Most of these (62%) occurred before the start of the COVID-19 vaccination campaign in nursing homes in December 2020.

Experience with a previous cluster was found to be a protective factor for the number of cases, hospital admissions and deaths occurring within subsequent clusters. A plausible explanation is the natural immunity developed after an infection, although the variable length of protection through antibodies permits the possibility of reinfection [16]. Another explanation might be that the frailest people living in the nursing home died during the first cluster, and therefore there were fewer hospital admissions and deaths during the next cluster. Third, the knowledge about IPC measures and outbreak management gained during the first cluster improved the implementation of these during subsequent clusters, which could be handled better or faster.

The first vaccination campaign organized in nursing homes was another protective factor for the number of cases, hospital admissions and deaths occurring within a cluster. In Belgium, 89.4% and 76.8% of nursing home residents and staff, respectively, received the primary course of the Comirnaty vaccine between 28 December 2020 and 24 March 2021 [5]. This vaccine has been shown to be highly immunogenic and effective to control outbreaks in nursing homes [7,17]. Moreover, a single dose of vaccination was associated with lower nasopharyngeal viral load than detected in absence of vaccination [18]. Following the univariate analyses, the vaccination rates only had a limited influence on the clusters still occurring after the start of the vaccination (Appendix A). In Belgium, the vaccination coverage among nursing home residents and staff was high after the primary course of the COVID-19 vaccination [5]. The difference between the nursing homes may not have been high enough to obtain a significant result.

During the year following the vaccination campaign for the primary course, 858 clusters were still detected. Although the number of clusters and the median AR among residents and staff, the median CHR and the median CFR dropped after the campaign, outbreaks with high AR resulting in high morbidity and high mortality among residents continued to occur in nursing homes in Belgium and other European countries (EU/EEA) [19,20,21]. The risk for COVID-19 infection in fully vaccinated residents and staff was much lower compared to unvaccinated residents and staff, but this was no longer significant in outbreaks with an AR of 20% or higher [19]. The most plausible explanation is a high force of infection because of exposure to large viral loads, which might overcome the immune protection [19]. This underlines the importance of early detection and rapid containment of outbreaks in nursing homes, to limit a broad circulation of SARS-CoV-2, through rapid testing of all residents and staff and ensuring strict infection prevention and control measures, even after vaccination [19,22].

The region where the nursing home was located was associated with the number of hospital admissions and deaths in a cluster. An outbreak investigation in a Dutch nursing home suggested a widespread regional circulation of the virus in the weeks before the outbreak as source of the outbreak [23], which can explain the regional influence. Larger nursing homes were associated with more cases and longer lasting clusters, while private (both for profit and nonprofit) nursing homes were associated with shorter lasting clusters.

Institutional characteristics associated with higher infection rates among nursing home residents remain unclear, and findings are contradictory. A large-scale study conducted in England showed reduced infection rates among residents and staff (and smaller outbreaks) related to factors as staff-to-bed ratio and paid sick leave. Factors such as staff caring for both infected and uninfected residents, new admissions, frequent employment of agency nurses, experiencing difficulties in isolating residents and for-profit institutions were linked to higher infection rates among residents and staff and larger outbreaks [24]. A large-scale regional study in Belgium showed that higher infection rates among residents were linked to an increased infection rate among nursing home staff members and a higher fraction of beds for individuals who were heavily dependent. However, no association with the size or type of nursing home, mean age of residents/staff or the proportion of asymptomatic positive tested cases was found [25]. Other large-scale studies, conducted in the United States, Scotland and Canada did not find such associations [26,27,28]. Since the provision and organizational structure of the nursing home setting can vary between countries and regions, comparing them with other geographical regions is challenging [1,25].

The natural experiment of Reilly et al. showed that clusters burned out when approximately 60% of the residents were infected, and results did not show a resurgence of the COVID-19 cluster early in the pandemic [29]. For more than half of the nursing homes, at least two clusters have been detected since 22 June 2020 in our study, with a median AR of 9% among residents, which might be too low to achieve herd immunity. In contrast to the study by Reilly et al., our study period was long enough to see the emergence of different variants of concern, which may also have had an impact on cluster recurrence.

Different kinds of IPC measures have been implemented at regional and local level (e.g., quarantine measures, cohorting of (possible) cases among residents and staff, use of additional PPE, preventive and rapid testing, ventilation, education support) [1]. The median AR, the median CHR and the median CFR after the vaccination campaign in Belgium were lower compared to the results in 10 EU/EEA countries [19]. The observed low median AR among residents and staff, both before and after the vaccination campaign, demonstrates a hopeful message: that the applied IPC measures may be effective in preventing widespread SARS-CoV-2 transmission in nursing homes [22].

### 4.2. Strengths and Limitations

Our study has several strengths. It describes and explores a unique time-series data collection in a frail nursing home population on whom the COVID-19 epidemic had a major impact. Large-scale surveillance data representing 99% of all nursing homes located in Belgium were analysed in detail. Additionally, we could observe the role of nursing home characteristics for a period of 20 months and could evaluate the impact of the COVID-19 primary vaccination course.

However, the absolute numbers of clusters should be interpreted with caution, as no genome sequencing or results about viral load are available. In the absence of sequencing data and a cluster investigation, we are unable to confirm that cases in a cluster are epidemiologically linked [20]. Additional information about frailty, comorbidities, age, the variant causing the cluster, IPC measures taken to control the cluster, booster vaccination, serology (immunosenescence), reinfection and symptomatic/asymptomatic cases are missing. Lacking more information about these clusters, for example, availability of personal protection equipment and hand sanitizers, infection and prevention control training, ventilation characteristics, cleaning procedures, and mitigation measures such as application of cohorting of wards and/or staff across the different waves, might interfere the extrapolation of the results. Moreover, the first wave was not included in the analysis, which could be seen as a limitation. A confirmed case has been defined as either having a PCR or a rapid antigen positive test, with no distinction between both. A PCR test is more sensitive than a rapid antigen test [30], which can be seen as a limitation of the study. Finally, as part-time employees and care staff members were not identifiable, we chose to not include the staff-to-bed ratio as a possible influencing factor. Measures that compute nurse staffing levels by dividing the total number of full-time equivalent nursing staff members by patient days or by the total number of beds are too crude. Patient-to-nurse ratio or nursing hours per patient day are better indicators to estimate nurse staffing levels and their impact on the quality of care [31].

## 5. Conclusions

Our results indicate that more than 2000 clusters were detected in Belgian nursing home during the study period, of which 40% still occurred after the start of the vaccination campaign. The majority of large clusters (≥10 confirmed cases among residents only) were detected before the start of the COVID-19 vaccination campaign. Our analysis showed that outbreak history and the vaccination campaign were associated with a decrease in the number of cases, hospital admissions and deaths among residents. We therefore recommend continuing to implement targeted interventions such as vaccination, broad testing upon signs of a cluster and the immediate implementation of additional IPC measures in case of a cluster in a nursing home.

## Figures and Tables

**Figure 1 viruses-15-00232-f001:**
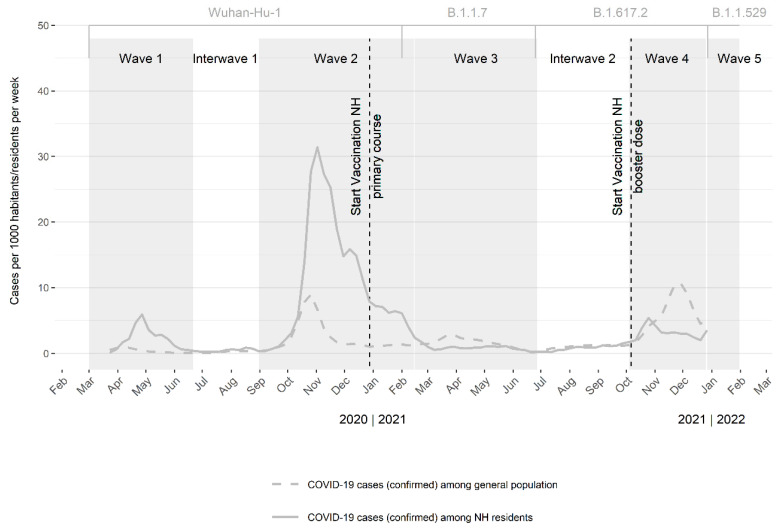
Confirmed cases per 1000 nursing home (NH) residents and per 1000 inhabitants, per week, in Belgium. A two-week moving average is presented from 17 March 2020–2 January 2022. The nursing home residents are included in the confirmed cases among the general population. Grey boxes indicate the COVID-19 (coronavirus disease 2019) waves. Grey brackets indicate the dominant variant. The dotted lines indicate the COVID-19 vaccination campaign (primary course) and the booster administration in nursing homes. Wuhan-Hu-1: Severe acute respiratory syndrome coronavirus 2 (wild type); B.1.1.7: Alpha variant; B.1.617.2: Delta variant; B.1.1.529: Omicron variant; NH: nursing home.

**Figure 2 viruses-15-00232-f002:**
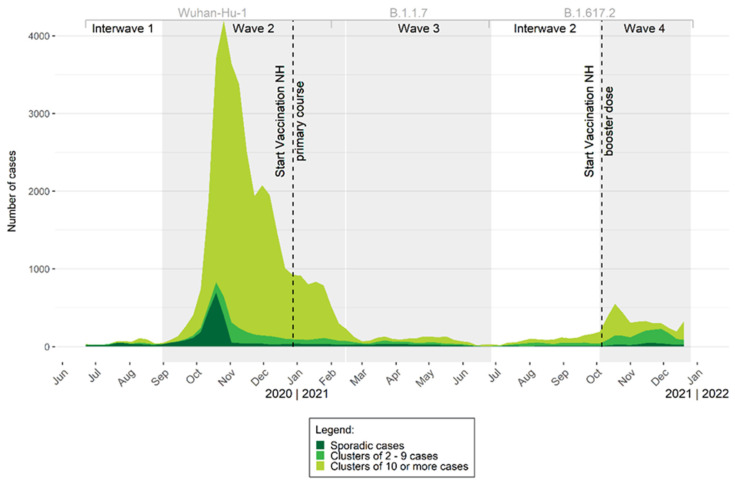
Evolution of the number of confirmed COVID-19 cases among nursing home (NH) residents by week in Belgium, from 22 June 2020 to 2 January 2022. Grey boxes indicate the second, third and fourth COVID-19 wave (from left to right). Dotted lines indicate the starts of the primary course vaccination campaign and of the booster vaccination. Wuhan-Hu-1: Severe acute respiratory syndrome coronavirus 2 (wild type); B.1.1.7: Alpha variant; B.1.617.2: Delta variant; NH: nursing home.

**Table 1 viruses-15-00232-t001:** Definitions applied to calculate possible clusters.

(Possible) Cluster *	Two or more CONFIRMED COVID-19 cases among residents were reported within one week (7 days) in the institution (same accreditation number)
Closure of a cluster	No new CONFIRMED COVID-19 cases were reported within 14 days (or if the institution stopped reporting: the cluster is automatically closed if there is no new information one month after the last reported case)
Start date of the cluster	Date where the first CONFIRMED COVID-19 case among residents of the cluster was registered
New cluster	The start date of the cluster falls within the reference week
Ongoing cluster	The cluster existed already before the reference week and is not closed yet in the reference week

* The authors prefer to speak about “possible clusters” because clusters were theoretically detected by applying the cluster definition on the surveillance data. An outbreak investigation would be needed to really confirm the cluster, by assessing the probability of the epidemiologic link between the cases. For readability, the word ‘cluster’ will be used from this point, and should be read as ‘possible cluster’.

**Table 2 viruses-15-00232-t002:** Nursing homes participating in the COVID-19 surveillances, overall and by regional health authorities, on 22 June 2020.

	Nursing Homes n	Beds Median (IQR)	Staff Members Median (IQR)
Belgium	1533	91 (65–120)	63 (38–92)
Brussels	142	114 (76–150)	61 (38–84)
Flanders	813	94 (68–122)	70 (42–102)
Wallonia	578	85 (58–110)	57 (35–83)

IQR: interquartile range.

**Table 3 viruses-15-00232-t003:** Cluster characteristics (size (number of COVID-19 confirmed cases), duration (days)), number of hospital admissions and deaths, per population (total, residents, and staff) and vaccination period expressed as median and interquartile range, recorded in Belgian nursing homes between 22 June 2020 and 2 January 2022.

Period	Total		Residents						Staff	
	Median Size	Median Du-Ration (Days)	Median Size	Median AR	Median of Number of Hospital Admissions	Median CHR	Median of Number of Deaths	Median CFR	Median Size	Median AR
Before start vaccination (since 22 June) (*n* = 1381)	16 (5–43)	27 (19–41)	12 (4–33)	13.3% (4.3–39.5%)	1 (0–3)	2.7% (0–12.5%)	1 (0–5)	9% (0–20%)	3 (0–10)	3.9% (0–13.7%)
After start vaccination (*n* = 858)	8 (4–16)	22 (18–29)	5 (2–11)	4.8% (2.7–12.2%)	0 (0–1)	0% (0–7.7%)	0 (0–1)	0% (0–8.3%)	2 (0–5)	2.5% (0–6.5%)
After start first dose (*n* = 102)	8 (3–21)	23 (18–31)	6 (2–15)	5.6% (2.6–17.9%)	0 (0–1)	0% (0–7.7%)	1 (0–2)	8.2% (0–25%)	1 (0–5)	1.2% (0–6.7%)
After start second dose (*n* = 756)	7 (4– 15)	22 (18–29)	5 (2–11)	4.8% (2.7–11.1%)	0 (0–1)	0% (0–7.3%)	0 (0–1)	0% (0–5.3%)	2 (0–5)	2.7% (0–6.5%)
Entire study period (*n* = 2239)	11 (4–30)	25 (18–36)	8 (3–22)	8.8% (3.4–26.7%)	0 (0–2)	0% (0–11%)	0 (0–3)	0% (0–18%)	2 (0–7)	3.2% (0–10.6%)

AR: attack rate; CHR: case hospitalization rate; CFR: case fatality rate, IQR: interquartile range.

**Table 4 viruses-15-00232-t004:** Comparison of cluster characteristics (size, duration, number of hospital admissions and deaths among residents) with nursing home features (size, ownership, region, wave, previous cluster experience, vaccination status), using Mann–Whitney U test/Kruskal–Wallis, between 22 June 2020 and 2 January 2022, Belgium.

	Group (*n*)	Mean Score	Mann–Whitney U Test	Kruskal–Wallis
			*p*-Value	*p*-Value
Cluster Size (Number of Cases Among Residents)				
Size of the nursing home	Small (918)	1146.4	0.82	
	Large (1366)	1139.9		
Ownership of the nursing home	Public (668)	1118.1		0.75
	Private nonprofit (847)	1105.3		
	Private for-profit (694)	1092.0		
Region	Flanders (1108)	1153.9		0.03
	Brussels (229)	1032.5		
	Wallonia (947)	1155.7		
Wave	Interwave 1 (70)	665.1		<0.001
	2nd wave (1406)	1323.3		
	3th wave (187)	741.2		
	Interwave 2 (131)	946.8		
	4th wave (441)	870.0		
	5th wave (49)	1144.6		
Previous cluster	No (292)	1420.6	<0.001	
	Yes (1992)	1101.7		
Start primary course vaccination	Vaccination not started (1419)	1293.1		<0.001
	First dose given (104)	1014.0		
	Second dose given (761)	879.2		
Vaccination rate staff (2 doses primary course) ≥ guideline of European Commission*	No (408)	384.6		0.23
	Yes (343)	365.8		
Vaccination rate residents (2 doses primary course) ≥ guideline of European Commission *	No (103)	340.0		0.04
	Yes (657)	386.8		
Duration of clusters (days)				
Size of the nursing home	Small (918)	1093.1	0.004	
	Large (1366)	1175.7		
Ownership of the nursing home	Public (668)	1139.2		0.24
	Private nonprofit(847)	1094.2		
	Private for-profit (694)	1085.3		
Region	Flanders (1108)	1145.7		0.14
	Brussels (229)	1063.2		
	Wallonia (947)	1158.0		
Wave	Interwave 1 (70)	655.1		<0.001
	2nd wave (1406)	1226.5		
	3th wave (187)	837.0		
	Interwave 2 (131)	1021.2		
	4th wave (441)	1076.2		
	5th wave (49)	1515.0		
Previous cluster	No (292)	1227.5	0.02	
	Yes (1992)	1130.0		
Start primary course vaccination	Vaccination not started (1419)	1204.6		<0.001
	First dose given (104)	993.7		
	Second dose given (761)	1047.1		
Vaccination rate staff (2 doses primary course) ≥ guideline of European Commission *	No (408)	385.8		0.18
	Yes (343)	364.3		
Vaccination rate residents (2 doses primary course) ≥ guideline of European Commission *	No (103)	399.1		0.35
	Yes (657)	377.6		
Number of hospital admissions among residents				
Size of the nursing home	Small (918)	1165.2	0.15	
	Large (1366)	1127.3		
Ownership of the nursing home	Public (668)	1124.3		0.53
	Private nonprofit (847)	1089.7		
	Private for-profit (694)	1105.1		
Region	Flanders (1108)	1117.6		0.0006
	Brussels (229)	1045.3		
	Wallonia (947)	1195.1		
Wave	Interwave 1 (70)	914.2		<0.001
	2nd wave (1406)	1301.7		
	3th wave (187)	878.9		
	Interwave 2 (131)	938.6		
	4th wave (441)	886.5		
	5th wave (49)	756.6		
Previous cluster	No (292)	1346.9	<0.001	
	Yes (1992)	1112.5		
Start primary course vaccination	Vaccination not started (1419)	1287.7		<0.001
	First dose given (104)	983.9		
	Second dose given (761)	893.4		
Vaccination rate staff (2 doses primary course) ≥Guideline of European Commission*	No (408)	388.6		0.03
	Yes (343)	361.0		
Vaccination rate residents (2 doses primary course) ≥ guideline of European Commission*	No (103)	405.7		0.13
	Yes (657)	376.6		
Number of deaths among residents				
Size of the nursing home	Small (918)	1148.1	0.72	
	Large (1366)	1138.7		
Ownership of the nursing home	Public (668)	1125.6		0.0002
	Private nonprofit(847)	1151.2		
	Private for-profit (694)	1028.7		
Region	Flanders (1108)	1254.2		<0.001
	Brussels (229)	1030.0		
	Wallonia (947)	1039.0		
Wave	Interwave 1 (70)	877.2		<0.001
	2nd wave (1406)	1339.0		
	3th wave (187)	835.0		
	Interwave 2 (131)	821.4		
	4th wave (441)	833.3		
	5th wave (49)	697.9		
Previous cluster	No (292)	1323.5	<0.001	
	Yes (1992)	1116.0		
Start primary course vaccination	Vaccination not started (1419)	1313.8		<0.001
	First dose given (104)	1152.3		
	Second dose given (761)	821.8		
Vaccination rate staff (2 doses primary course) ≥ guideline of European Commission *	No (408)	367.2		0.14
	Yes (343)	386.4		
Vaccination rate residents (2 doses primary course) ≥ guideline of European Commission *	No (103)	393.1		0.44
	Yes (657)	378.5		

* Only for clusters occurring after the start of the vaccination campaign.

**Table 5 viruses-15-00232-t005:** Multivariate analysis of factors influencing the nursing home cluster characteristics (size, hospital admissions, deaths, duration) (negative binomial regression) for the period from 22 June 2020 to 2 January 2022, Belgium.

	Cluster Size		Hospital Admissions		Deaths		Cluster Duration (Days)	
	Estimate (Wald 95% CI)	IRR (95% CI)	Estimate (Wald 95% CI)	IRR (95% CI)	Estimate (Wald 95% CI)	IRR (95% CI)	Estimate (Wald 95% CI)	IRR (95% CI)
Intercept	3.27 *** (3.14–3.40)		0.95 *** (0.75–1.16)		1.04 *** (0.80–1.28)		3.51 *** (3.42–3.61)	
Ownership								
*Public*					Ref.	Ref.	Ref.	Ref.
*Private nonprofit*					−0.10 (−0.29–0.08)	0.90 (0.75–1.08)	−0.09 ** (−0.16–−0.03)	0.91 ** (0.85–0.97)
*Private for-profit*					−0.08 (−0.28–0.11)	0.92 (0.76–1.12)	−0.08 ** (−0.15–−0.01)	0.93 ** (0.86–0.99)
Size								
*Small (p < 50)*	Ref.	Ref.					Ref.	Ref.
*Large (p > 50)*	0.23 *** (0.14–0.32)	1.26 *** (1.15–1.37)					0.12 *** (0.06–0.18)	1.13 *** (1.07–1.20)
Region								
*Flanders*	Ref.	Ref.	Ref.		Ref.	Ref.		
*Brussels*	−0.11 (−0.26–0.03)	0.89 (0.77–1.03)	−0.15 (−0.41–0.12)	0.86 (0.66–1.13)	−0.13 (−0.41–0.15)	0.88 (0.66–1.16)		
*Wallonia*	0.05 (−0.04–0.14)	1.05 (0.96–1.15)	0.32 ** (0.16–0.47)	1.37 *** (1.17–1.60)	0.34 ** (0.18–0.51)	1.41 ** (1.19–1.67)		
Previous cluster								
*No*	Ref.	Ref.	Ref.	Ref.	Ref.	Ref.	Ref.	Ref.
*Yes*	−0.34 *** (−0.47–−0.21)	0.71 *** (0.63–0.81)	−0.36 *** (−0.57–−0.15)	0.70 *** (0.57–0.86)	−0.40 *** (−0.61–−0.18)	0.67 *** (0.54–0.84)	−0.05 (−0.14–0.04)	0.95 (0.87–1.04)
Start primary course vaccination								
*Vaccination not started*	Ref.	Ref.	Ref.	Ref.	Ref.	Ref.	Ref.	Ref.
*First dose given*	−0.54 *** (−0.75–−0.33)	0.58 *** (0.47–0.72)	−0.73 *** (−1.10–−0.36)	0.48 *** (0.33–0.70)	−0.75 ** (−1.12–−0.38)	0.47 *** (0.33–0.69)	−0.21 ** (−0.34–−0.07)	0.81 ** (0.71–0.93)
*Second dose given*	−0.92 *** (−1.0–−0.83)	0.40 *** (0.36–0.44)	−1.36 *** (−1.53–−1.19)	0.26 *** (0.22–0.31)	−1.35 *** (−1.52–1.17)	0.26 *** (0.22–0.31)	0.28 *** (0.22–0.34)	1.32 *** (1.24–1.40)
Dispersion Parameter	0.98 (0.93–1.04)		2.28 (2.06–2.51)		2.28 (2.06–2.53)		0.40 (0.38–0.43)	
Log. Lik.	−8743.85		−3579.25		−3466.07		−9746.22	
AIC	17,503.69		7172.49		6950.14		19,508.43	

** *p* < 0.05, *** *p* < 0.001. IRR: incidence rate ratio; 95% CI: 95% confidence interval; Log. Lik.: log-likelihood; AIC: Akaike information criterion. Cluster size: number of cases among nursing home residents.

## Data Availability

The datasets used and/or analysed during the current study are available from the corresponding author on reasonable request.

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
