# Peer review of "COVID-19 Clusters in Belgian Nursing Homes: Impact of Facility Characteristics and Vaccination on Cluster Occurrence, Duration and Severity"

_viruses, 2023, doi:10.3390/v15010232_

Round 1

Reviewer 1 Report

Dequecker and colleagues present an analysis on outbreaks of COVID-19 in Belgian nursing homes. The authors report that most COVID-19 clusters occurred before the start of vaccination. Previous cluster experience and having started vaccination were 29 protective factors. The paper is well-written, and the statistical methods are appropriate. I therefore only have a few minor comments.

1.     I kindly request you to reduce the number of abbreviations throughout the paper. Please write out abbreviations such as NH, LCTF and W1 through W5. In my view, this would improve the readability of the paper.

2.     On page 11 there is a summary of IRR values. Please add the 95% confidence interval to these values as the 95% confidence interval can be used to assess statistical significance and precision of the IRR.

3.     In Table 3 please remove statistical data such as the U, Z, chi square and degrees of freedom as these data do, in my view, not add too much information. I would recommend reporting the cluster size as a median and interquartile range rather than a mean which assumes a normal distribution.

4.     The authors define a confirmed case as either having a PCR or a rapid antigen positive test. A PCR test is more sensitive than a rapid antigen test. I recommend to add this information as a possible limitation to the paper.

Reviewer 2 Report

In this study, the authors assessed the impact of facility characteristics and vaccination on cluster occurrence, duration, and severity among nursing homes in Belgium. It was an interesting topic that could evaluate the effect of the COVID-19 primary vaccination course among the targeted population. Some minor suggestions should be mentioned below:

1) There were 2,239 clusters identified in this study, more descriptions for these clusters should be introduced that influence the extrapolation of the conclusion. 

2) The data extraction process needs to be described in detail
